# Microsecond MD Simulations to Explore the Structural and Energetic Differences between the Human RXRα-PPARγ vs. RXRα-PPARγ-DNA

**DOI:** 10.3390/molecules27185778

**Published:** 2022-09-07

**Authors:** Faizul Azam, Martiniano Bello

**Affiliations:** 1Department of Pharmaceutical Chemistry and Pharmacognosy, Unaizah College of Pharmacy, Qassim University, Unaizah 51911, Saudi Arabia; 2Laboratorio de Diseño y Desarrollo de Nuevos Fármacos e Innovación Biotecnológica, Escuela Superior de Medicina, Instituto Politécnico Nacional, Plan de San Luis y Diaz Mirón, s/n, Col. Casco de Santo Tomas, Ciudad de México 11340, Mexico

**Keywords:** RXRα, PPARγ, MD simulations, MMGBSA, binding free energy

## Abstract

The heterodimeric complex between retinoic X receptor alpha (RXRα) and peroxisome proliferator-activated receptor gamma (PPARγ) is one of the most important and predominant regulatory systems, controlling lipid metabolism by binding to specific DNA promoter regions. X-ray and molecular dynamics (MD) simulations have revealed the average conformation adopted by the RXRα-PPARγ heterodimer bound to DNA, providing information about how multiple domains communicate to regulate receptor properties. However, knowledge of the energetic basis of the protein-ligand and protein-protein interactions is still lacking. Here we explore the structural and energetic mechanism of RXRα-PPARγ heterodimer bound or unbound to DNA and forming complex with co-crystallized ligands (rosiglitazone and 9-*cis*-retinoic acid) through microsecond MD simulations, molecular mechanics generalized Born surface area binding free energy calculations, principal component analysis, the free energy landscape, and correlated motion analysis. Our results suggest that DNA binding alters correlated motions and conformational mobility within RXRα–PPARγ system that impact the dimerization and the binding affinity on both receptors. Intradomain correlated motions denotes a stronger correlation map for RXRα-PPARγ-DNA than RXRα-PPARγ, involving residues at the ligand binding site. In addition, our results also corroborated the greater role of PPARγ in regulation of the free and bound DNA state.

## 1. Introduction

Nuclear receptors (NRs) are a large group of multidomain proteins that regulate the transcription of numerous genes in humans by attaching to specific sequences and regulating gene expression [1,2,3,4]. Peroxisome proliferator-activated receptors (PPARs) are ligand-activated transcription factors belonging to the NR superfamily [5]. There are three subtypes of PPARs-α, γ, and δ-each with distinct tissue distribution [6,7,8,9]. PPARγ is found in the liver, kidney, muscle, adipose tissue, and heart, and it is essential in adipogenesis. Its overexpression enhances adipogenesis in vitro [10]. PPARγ plays a critical role in regulating glucose and lipid metabolism; therefore, it has been associated with diabetes, obesity, and cardiovascular diseases [11]. Thiazolidinedione (TZD) compounds are a class of PPARγ ligands that improve insulin resistance by promoting adipocyte differentiation via PPARγ activation in adipocytes with a high capability to utilize glucose [12,13]. Rosiglitazone (RGZ) is a type of TZD that has been commercialized due to its euglycemic effects [14].

Retinoic X receptors (RXRs) are also part of the NR superfamily and form obligate heterodimers with PPARs [15]. These receptors bind the vitamin A metabolite 9-*cis*-retinoic acid (9CR) [16]. There are three subtypes of RXRs-α, γ, and δ- which bind to DNA and activate transcription in response to 9CR. RXRs also form heterodimer complexes with other class 1 NRs, such as thyroid hormone, retinoic acid, and vitamin D receptors [17]. Some of the permissive RXR heterodimers are those formed with PPARs, so they can be activated by 9CR. Modulating the permissive heterodimers by cognate ligands offers an additional level of regulation [17]. Topologically, these NRs consist of three major domains: (1) an N-terminal domain, also termed the A/B region; (2) a small, well-conserved DNA binding domain (DBD) containing two zinc fingers that recognize specific DNA sequences, alongside a flexible hinge that connects the DBD to the ligand-binding domain (LBD); and (3) an LBD containing the activation function-2 (AF-2) surface at its C-terminus and a dimerization interface (Figure 1) [18,19]. The LBD of PPARs comprises 13 α-helices and four short β-strands [20]. Helices H2 and H3 form a very flexible loop localized at the entrance of the ligand-binding site [21]. The LBD of RXRα comprises 11 α-helices and two short β-strands [22]. The RXRα–PPARγ system regulates gene transcription by binding to specific DNA promoter regions [23]. In the apo state, H12 present in LBD exhibits highly conformational mobility ranging from active to inactive states. Upon ligand binding, the mobility H12 can be stabilized by altering the region of LBD containing AF-2 [24].

Structural data collected by X-ray crystallography, electron microscopy, and small-angle X-ray scattering (SAXS) have revealed that the binding of either a coactivator or a corepressor switches the heterodimer complex from the active to the inactive state. In the active state, the heterodimer is coupled to activating ligands, and the coactivators bind DNA and activate gene expression. In contrast, in the inactive state, the activating ligands and the coactivators are substituted by corepressors [27,28,29]. Recent evidence has shown that the structural mobility of LBDs is impacted by their interaction with DNA [30], and ligand-induced activation is dependent on the anchoring DNA sequence [31,32]. Molecular dynamics (MD) simulations of the RXRα-PPARδ heterodimer showed that the mobility of H10 and H11 (H10/11) of PPARα is affected by RXRα ligand binding, indicating that these protein regions are involved in allosteric communication [33]. Amber-Vitos et al. performed MD simulations using active and modeled inactive RXRα-PPARδ-DNA systems [34]. They found that in the forced inactive state, the two LBDs, the hinge region, and the two DBD regions are much more rigid. Recently, dynamic simulation in conjunction with analysis of correlated motions and using the active RXRα–PPARγ–DNA system identified distant correlated motions. Ricci et al. performed MD simulations and identified changes at the nanosecond timescale, namely a flux of correlated motions from DNA directed to the LBDs, mediated by the DBD of PPARγ, but without the participation of the RXRα DBD [35]. This finding aligns with hydrogen-deuterium exchange experiments using the vitamin D receptor (VDR)-RXR complex [30]. Although there has been support for communication between DBDs, LBDs, and DNA, this communication has also been observed in the non-permissive form of RXR–retinoic acid receptor (RAR) heterodimers [36]. These structural data and MD simulations support the presence of synergism and communication between DBDs, LBDs, and DNA of the ternary RXRα-PPARγ-DNA system. Understanding these phenomena, together with an energetic study (still lacking), could increase knowledge about this system. In this study, we explore the structural and energetic basis of the active RXRα-PPARγ, and RXRα-PPARγ-DNA systems in which the heterodimer is forming complex with 9CR and RGZ through triplicate microsecond MD simulations, followed by comparative analyses in terms of binding free energies using the molecular mechanics generalized Born surface area (MMGBSA) approach, principal component analysis (PCA), the free energy landscape (FEL), and correlated motion analyses. Our results shed light on the dynamic and energetic aspects of the allosteric communication of the RXRα-PPARγ and RXRα-PPARγ-DNA systems.

## 2. Results

### 2.1. Stability of the Simulated RXRα-PPARγ-DNA and RXRα-PPARγ Systems

The root-mean-square deviation (RMSD), radius of gyration (RoG), and solvent-accessible surface area (SASA) plots illustrate the convergence of the RXRα-PPARγ-DNA and RXRα-PPARγ systems when bound to RZG and 9CR. Triplicate simulations of the RXRα-PPARγ-DNA system showed that the RMSD (Appendix A), RoG (Appendix A) and SASA (Appendix A) for backbone atoms first achieve equilibrium between 0.1 to 0.2 µs and then again at 0.6 µs, with RoG and SASA values that oscillated between 27.8–28.2 Å and 3650–3750 Å^2^, respectively. The RMSD (Appendix A), RoG (Appendix A) and SASA (Appendix A) for the RXRα-PPARγ system also achieve constant values between 0.1–0.2 and then at 0.6 µs, with RoG and SASA values that fluctuated between 28.2–28.6 Å and 3550–3700 Å^2^, respectively. Therefore, we considered the last 0.4 µs as the equilibrated part of each simulation.

### 2.2. RMSF Analysis of the Simulated RXRα-PPARγ-DNA and RXRα-PPARγ Systems

RMSF analysis over the equilibrated simulation time shows that RXRα forming part of the RXRα-PPARγ system exhibits higher mobility that in RXRα-PPARγ-DNA (Appendix A), particularly at DBD (residues 132–168 and 170–188), the hinge (residues 210–222), H2 (242–261), and part of H10/H11 and AF-2 (residues 438–448). PPARγ in the RXRα–PPARγ system display higher mobility at DBD (residues 108–173), and the hinge (residues 175–189) than in the RXRα-PPARγ-DNA system (Appendix A). In contrast, PPARγ in the RXRα-PPARγ-DNA system display higher mobility at the hinge (residues 193–205), H2’ and H2’-H3 loop (251–276).

### 2.3. Ligand Interactions of the PPARγ-RXRα-DNA System 

#### 2.3.1. RZG at the PPARγ Binding Pocket

Clustering analysis over the equilibrated simulation time allowed us to obtain the first, second, and third populated conformers (Appendix A). These three conformers concentrated the highest percentage of conformations for RXRα-PPARγ-DNA and RXRα-PPARγ systems (Appendix A), providing relevant information about the representative interactions during MD simulations. Overlapping of the three conformers illustrates that the regions with significant structural differences were at DBD (Appendix A) or hinge (Appendix A) of PPARγ within RXRα-PPARγ or RXRα-PPARγ-DNA systems. The first populated conformation of the RXRα-PPARγ-DNA system showed that RGZ forms hydrogen bonds (HBs) through its TZD headgroup with the sidechain and polar backbone atoms of E343. The TZD headgroup also makes hydrophobic and polar contacts with P227, R288, S289, M329, L330, L333, and S342. The pyridyl tail is stabilized by F282, C285, Q286, I326, I456, L469, and Y473 (Figure 2A). The second populated conformer showed that the TZD headgroup forms HBs with the sidechain and polar backbone atoms of E343. The portion containing TZD headgroup also makes hydrophobic and polar contacts with C285, Q286, S289, I326, M329, L330, L333, S342, and M364. The pyridyl group is stabilized by Q283, Y327, P359, F360, F363, L452, L453, H449, and I456 (Figure 2C). Among these residues, H449 forms one HB with the pyridyl group.

The third populated complex also showed the HBs between the TZD headgroup and the sidechain and polar backbone atoms of E343.The portion containing the TZD headgroup makes hydrophobic contacts with C285, S289, M329, L330, L333, L340, I341, and S342. The pyridyl group is bound by F282, Q283, Q286, I326, Y327, F360, F363, and H449 (Figure 2E). Comparison among the three conformers showed that despite of the differences in the three cases the stabilization of the TZD headgroup was through HBs with E343 (Figure 3).

Based on MD simulations, HBs are mainly between the TZD headgroup and E343 localized at β2–β4, whereas in different PPARγ-RXRα structures (PDB entries: 1FM6 [22], 3DZY [25], and 5JI0) (Appendix A), the TZD headgroup did not make HBs with E343. With respect to hydrophobic interactions, most of the interactions localized at H3 (F282, C285, Q286, and S289), H4/5 (Y327 and L330), β2–β4 (I341), H6 (L353), H7 (M364), and H10/11 (H449), observed in the co-crystallized PPARγ-RXRα-DNA complex (Appendix A) and in other PPARγ-RXRα structures: 1FM6 [22] (Appendix A) and 5JI0 (Appendix A), are still present during simulations. 

#### 2.3.2. 9CR at the RXRα Binding Pocket

In the first populated RXRα-PPARγ-DNA system, 9CR makes a series of van der Waals contacts with the ligand binding site of RXRα. The carboxylate of 9CR makes polar interactions with N262. The triene portion and the β ionone establish interactions with P264, T266, C269, A272, Q275, L276, T278, W305, L451, and M454 (Figure 2B). In the second populated conformation, the carboxylate of 9CR forms one HB with backbone atoms of N262. The hydrophobic regions establish interactions with P264, T266, C269, A272, Q275, L276, T278, L279, W305, L309, A327, L451, and M454 (Figure 2D). In the third conformation, the carboxylate of 9CR forms one HB with the side chain of S260. The nonpolar region establishes interactions with N262, P264, T266, C269, A272, Q275, L276, L279, W305, L309, and M454 (Figure 2F). Comparison with the crystallographic structure (Appendix A) showed that during MD simulations, the carboxylate of 9CR changes into different conformations that allow the formation of HBs with N262 and S260, not present in the RXRα-PPARγ-DNA structure [25] and in RXRα-PPARγ structures (PDB entries: 1FM6 [22] and 5JI0) (Appendix A). With respect to hydrophobic interactions, MD simulations revealed that some interactions along H3 (A272 and Q275), H4/H5 (W305 and L309), and the loop between H4/H5 and H6 (A327) observed in the RXRα-PPARγ-DNA structure [25] (Appendix A) and in different RXRα-PPARγ structures (Appendix A) are still present. In addition, in MD simulations there are some new hydrophobic interactions along H3 (P264, T266, C269, L276, and T278), and AF-2 (L451 and M454) not present in the crystallographic complex. 

### 2.4. Ligand Interactions of the RXRα-PPARγ System 

#### 2.4.1. RZG at the PPARγ Binding Pocket

The first populated conformer of the RXRα-PPARγ system showed the TZD headgroup forms one HB with Y473, a characteristic HB involved in stabilizing a full agonist in the active state of the RXRα-PPARγ system [20,37]. The TZD headgroup also forms contacts with F282, S289, I326, Y327, H449, L452, and L469. The pyridyl tail is bound by I262, C285, L330, and I341 (Figure 3A). These interactions are comparable to those observed in different crystallographic structure [25] of the RXRα-PPARγ-DNA complex (Appendix A). The second populated complex showed that the HBs occur between the TZD headgroup and the sidechains of H449, and Q286. The part of RZG containing TZD headgroup forms hydrophobic contacts with S289, H323, I326, Y327, F363, and Y473. The pyridyl tail is attached by C285, R288, L330, L340, I341, S342, and M364 (Figure 3C). From these residues, R288 forms a HB with polar atoms of pyridyl tail. The third populated complex illustrated that the TDZ group establishes HBs with Q286, S289, H323, and H449. The section containing TZD headgroup makes hydrophobic contacts with F282, I326, Y327, F363, and Y473. The pyridyl tail is attached by I262, G284, C285, I341, M348, and M364 (Figure 3E). Comparison of the three conformers with different RXRα-PPARγ structures (Appendix A) revealed that despite the differences in how the TZD headgroup is stabilized through HBs, there are interactions between the TZD headgroup and S289 (H3), H323 (H4/5), and Y473 (H7), in addition to other HBs with residues of H3 (Q286), H4/5 (Y327) and H10/11 (H449) not present in the crystallographic complex. These polar interactions are in line with those observed through MD simulations employing the monomeric state of PPARγ with different compounds containing the TZD group [38]. With respect to hydrophobic interactions, most of the interactions localized at H3 (F282, C285, Q286, R288, and S289), H4/5 (Y327 and L330), β2–β4 (341), H7 (M364), and H10/11 (H449) (Appendix A) detected in the co-crystallized complex [25] are even present in MD simulations. 

#### 2.4.2. 9CR at the RXRα Binding Pocket

In the first populated RXRα-PPARγ complex, 9CR is bound by a series of van der Waals interactions with V265, N267, F313, I324, V332, S336, A340, and V342 (Figure 3B). In the second conformation, 9CR establishes nonpolar interactions with V265, T266, L326, V332, V342, C432, L436, F439, and I447 (Figure 3D). In the third conformation, the ligand is bound by P264, V265, L326, V342, V349, I345, and C432. Comparison with the crystallographic structure [25] showed that during MD simulations, the carboxylate of 9CR changes into different conformations that do not allow the formation of HBs (Appendix A). 

MD simulations also showed that some hydrophobic interactions in H4/H5 (F313), the loop between H4/H5 and H6 (L326), H7 (V342 and I345), H10/11 (C432), and AF-2 (L436) observed in the co-crystallized complex (Appendix A) are still present. Taken together, the MD simulations revealed some new hydrophobic interactions in H3 (P264, V265, T266, and N267), the loop between H4/H5 and H6 (I324 and V332), H6 (S336 and A340), H7 (V349), H10/11 (F439), and AF-2 (I447) not present in the crystallographic complex. Comparison between the RXRα-PPARγ-DNA and RXRα-PPARγ systems during simulations showed that the latter share more interactions with those in the crystallographic complex than the PPARγ-RXRα-DNA system. This finding suggests that the ternary complex has a greater impact on 9CR binding to RXRα than RGZ binding to PPARγ.

### 2.5. The Protein–Protein Interactions of the RXRα-PPARγ-DNA and RXRα-PPARγ Systems

The heterodimeric interface of the RXRα-PPARγ-DNA and RXRα-PPARγ systems is stabilized by more protein–protein interactions than those present in the co-crystallized complex (Appendix A). Some similar interactions present during simulations and in the co-crystallized complex [25] are those involving the DBD, the hinge, H7, H8-H9 loop, and H10/11 of RXRα, and the DBD, β2–β4, H6, H7, H9, and H10/11 of PPARγ. In the RXRα-PPARγ system, the protein–protein interface is mostly stabilized by residues of the DBD and the hinge of RXRα, forming interactions with residues of the DBD, β2–β4, H2, H6, H7, and H10/11 of PPARγ. For the RXRα-PPARγ-DNA system, the protein–protein interface is mostly structured by residues of H7, H9, H8-H9 loop, and H10/11 of RXRα, forming interactions with residues of H7, H8, H8-H9 loop, H9, and H10/11 of PPARγ. Comparison between the RXRα-PPARγ-DNA and RXRα-PPARγ systems revealed more interactions for the latter system, mostly through the DBD and the hinge of RXRα with residues of the LBD of PPARγ. In contrast, in the RXRα-PPARγ-DNA system, most of the interactions are through the LBD of both receptors. These structural differences suggest different protein–protein affinity that would also impact the affinity of each ligand to bind to RXRα and PPARγ.

### 2.6. Binding Free Energy Calculations of Protein-Ligand Interactions Using MMGBSA

Binding free energy (*ΔG_MMGBSA_*) values determined with the MMGBSA approach for protein–ligand and protein–protein interactions are energetically favorable (Table 1). The changes in solvation free energy using GB (*ΔG_ele,sol_ + ΔG_npol,sol_*) revealed that RGZ shows a higher desolvation cost of binding than 9CR in RXRα-PPARγ-DNA and RXRα-PPARγ systems. In fact, this term contributes favorably for 9CR binding, in contrast to RZG binding, and this is appreciated for the two systems. Table 1 also shows that the protein–ligand interaction energy (*ΔE_vdw_*
*+*
*ΔE_ele_*) is energetically favorable for RZG binding in the RXRα-PPARγ-DNA and RXRα-PPARγ systems. Despite these differences, the *ΔG_MMGBSA_* values are thermodynamically favorable for RGZ and 9CR binding in the RXRα-PPARγ-DNA and RXRα-PPARγ systems. Interestingly, whereas the affinity of RGZ in both systems is similar, the affinity of 9CR in the RXRα-PPARγ-DNA system is higher in the RXRα-PPARγ system, suggesting that DNA binding by part of the RXRα-PPARγ system has greater impact on molecular recognition at the RXRα ligand binding site.

### 2.7. Binding Free Energy Calculations of Protein-Protein Interactions Using MMGBSA

Based on the changes in solvation free energy using GB, the RXRα-PPARγ system has a higher desolvation cost of protein–protein binding than the RXRα-PPARγ-DNA system, contributing unfavorably to the *ΔG_MMGBSA_* value (Table 1). The protein–protein interaction energy is energetically more favorable for the RXRα-PPARγ system than the RXRα-PPARγ-DNA system, consistently with the finding that there are more interactions for the RXRα-PPARγ system than the RXRα-PPARγ-DNA system (Appendix A). The affinity of the RXRα-PPARγ system is higher than the RXRα-PPARγ-DNA system, indicating that DNA binding by part of the RXRα-PPARγ system contributes to decrease the affinity between both receptors. Considering the differences in the protein–ligand and protein–protein affinity for the RXRα-PPARγ than RXRα-PPARγ-DNA systems, we can conclude that DNA binding by part of the RXRα-PPARγ system contributes to decrease the protein–protein affinity but increases the ligand affinity at the RXRα binding site.

### 2.8. Per-Residue Free Energy Decomposition 

#### 2.8.1. Per-Residue Free Energy Decomposition of the PPARγ_RZG_ Complex

Appendix A shows the energies for each residue participating in the protein–ligand interactions of the RXRα-PPARγ and RXRα-PPARγ-DNA systems. In the RXRα-PPARγ-DNA system, the major contributors to the binding affinity (*ΔG_MMGBSA_* ≥ 1.0 kcal) in the PPARγ_RZG_ complex are F282, C285, Q286, I326, L330, I341, E343, and His449 (Figure 4A). Of these residues, E343 forms HBs through its polar backbone atoms and side chain with the TZD headgroup (Figure 2A,C,E), whereas the rest stabilize RGZ through hydrophobic interactions. For the PPARγ_RZG_ complex in the RXRα-PPARγ-DNA system (Figure 4B), the major contributors to the affinity are C285, Q286, R288, I326, Y327, L330, I341, and H449 (Figure 3A,C,E). These residues mostly form hydrophobic interactions. Comparison of RGZ stabilization in both systems revealed that this ligand is better stabilized in the RXRα-PPARγ-DNA system, with more residues contributing to the binding affinity.

#### 2.8.2. Per-Residue Free Energy Decomposition of the RXRα_9CR_ Complex

In the RXRα-PPARγ-DNA system, the main sources of the binding free energy for RXRα_9CR_ are N262, P264, A272, Q275, L276, W305, L309, and L451 (Figure 4C). Each of these residues participates in hydrophobic interactions, except for N262, which forms HBs (Figure 2D). For RXRα_9CR_ in the RXRα-PPARγ system (Figure 4D), the major contributors to the affinity are N267, L326, V332, V342, and I345, which form hydrophobic interactions. Comparison of the interactions that mainly contribute to the binding free energy in RXRα_9CR_ indicated that 9CR is stabilized better in the RXRα-PPARγ-DNA system. Furthermore, the residues participating in ligand stabilization are different in the two systems, indicating that DNA binding to the RXRα-PPARγ system increases the stability of 9CR binding to RXRα.

#### 2.8.3. Per-Residue Free Energy Decomposition of the RXRα-PPARγ System

Appendix A lists the energies for each protein–protein interaction that maintains the RXRα-PPARγ and RXRα-PPARγ-DNA systems. In the RXRα-PPARγ system, the main sources of the binding free energy in the heterodimeric complex are Y169, R184, Y192, R209, N216, E219, E394, Y397, and R421 for RXRα (Figure 5A). Of these residues, Y169, R184, Y192, N216, and E219 are localized along the DBD, whereas E394 and R421 belong to H9 and H10/11, respectively. Regarding PPARγ, R153, D337, E351, E378, E396, Q430, K434, and Q437 are the main contributors to the complex affinity.

These residues are localized in the DBD (R153), β2–β4 (D337), H6 (E351), H7 (E378), H8-H9 loop (E396), and H10/11 (Q430, K434 and Q437).

In the RXRα-PPARγ-DNA system, the contributing residues are L196, R202, D214, E394, L419, L420, R421, S427, and K431 for RXRα (Figure 5B). These residues are located in the DBD (L196 and R202), the hinge (D214), H9 (E394), and H10/11 (L419, L420, R421, S427, and K431). With respect to PPARγ, D337, D396, L414, S429, K434, L436, M439, T447, Q451, and Y477 are the main residues that affect the affinity. These residues are located in β2–β4 (D337), H8-H9 loop (D396), H9 (L414), H10/11 (S429, K434, L436, M439, T447, and Q451) and H12 (Y477). 

Although both systems share several protein–protein interactions (Figure 5), those that contribute to *ΔG_MMGBSA_* are dissimilar, indicating that DNA binding by the RXRα-PPARγ system impacts the affinity as well as the type of interactions responsible for binding. Considering the protein regions that mostly contribute to *ΔG_MMGBSA_*, the structural domains that guide protein–protein recognition in the RXRα-PPARγ system are the DBD, H9, and H10/11 of RXRα, and the DBD, β2–β4, H6, H7, H8-H9 loop, and H10/11 of PPARγ. For the RXRα-PPARγ-DNA system, the domains are the DBD, the hinge, H9, and H10/11 of RXRα, and β2–β4, H8-H9 loop, H9, H10/11, and H12 of PPARγ.

### 2.9. PCA

PCA was carried out to extract the most important eigenvectors. For the RXRα-PPARγ and RXRα-PPARγ-DNA systems, the first two eigenvectors (PC1 and PC2) contain the largest eigenvalues, contributing 48.0% and 43.0%, respectively, to the total fluctuation of the two systems. PC1 versus PC2 contains the main conformational states sampled by simulations for the RXRα-PPARγ (Figure 6A) and RXRα-PPARγ-DNA (Figure 6B) systems. This comparison provided information about the conformational complexity by reconstructing the FEL (Figure 6C,D). Projection of the motion in the phase space along the first and second eigenvectors (PC2 versus PC1) for the RXRα-PPARγ system (Figure 6C) covers a larger region in the essential subspace than that of the RXRα-PPARγ-DNA system (Figure 6D), suggesting a larger conformational entropy for the RXRα-PPARγ system. Graphical representation of the projection along PC1 and PC2 shows that RXRα-PPARγ system systems display the highest collective motions along DBD and H2’-H3 loop of PPARγ and one small portion of DBD-hinge, H1-H2 loop and H11-H12 loop of RXRα (Figure 7A). For RXRα-PPARγ-DNA system, the highest fluctuation is along the hinge and H2’-H3 loop of PPARγ and H1-H2 loop of RXRα (Figure 7B). The H2’-H3 loop in RXRα-PPARγ system showed a sliding motion, strongly associated to PPARγ surface and in close proximity to form interactions with H6-H7 loop (Appendix A). In RXRα-PPARγ-DNA system, the H2’-H3 loop exhibited a flapping motion over the ligand binding site of LBD (Appendix A). On the other hand, the differences in mobility of H11-H12 loop, localized at the entrance of RXRα binding site, may be responsible for the variations in affinity for 9CR (Table 1). These results indicates that reduction in the conformational mobility of DBD of PPARγ seems to be the main responsible in the differences in the distribution along the essential subspace.

Figure 6C,D show the two-dimensional FEL of the two-member system utilizing the first two eigenvector projections as the reaction coordinates. The FEL indicates that although the RXRα-PPARγ-DNA system shows greater occupancy of the major stable conformational ensembles, two of the major conformational ensembles for the RXRα-PPARγ system (Figure 6C) are larger than those in the RXRα-PPARγ-DNA system. These findings indicate that the RXRα-PPARγ system samples more of the most thermostable conformations during the simulations. 

### 2.10. Dynamic Correlated Motions

We performed dynamic correlation analysis to explore the intradomain correlations within the PPARγ and RXRα and in the heterodimer state. The correlated motions among different regions of RXRα in the RXRα-PPARγ-DNA system are through hinge–H1, H1–H2, H2–H3, H3–H4/5, H4/5–H7, H7–H9, and H10/11–H12 (Figure 8). Of the residues participating in these correlated motions, C269 localized in H3 interacts directly with 9CR (Figure 2B,D,F), and E219 in the hinge participates in protein–protein interactions with Q163 at the DBD of PPARγ. 

For PPARγ in the RXRα-PPARγ-DNA system, the correlated motions are through DBD–hinge, hinge–H1, H1–H2, H2–H3, H3–H4/5, H4/5–H7, H7–H9, and H10/11–H12 (Figure 8). From the residues participating in these correlated motions, F363 and M364 localized in H7 interact directly with RZG (Figure 2C,E). R357 is in close contact with these residues; it has been reported to be important to regulate ligand dissociation [39]. Therefore, correlated motions involving R357 can be a mechanism of regulating ligand dissociation. H449 in H10/11 participates in protein–ligand interactions with RZG (Appendix A). This residue is near residues forming protein–protein interactions with H10/11 of RXRα, suggesting that correlated motions involving this residue might be a mechanism to regulate the protein–protein interface. Other residues (N312 and D313) in H4/5 whose correlations appeared in the RXRα-PPARγ-DNA system are in close proximity to the coactivator binding site. Therefore, this correlation could be linked to modulation of a coactivator. In addition, there is one interdomain correlated motion between R136 in the DBD of PPARγ and H449 in H12 of RXRα (data not showed), indicating correlated motions between the DBD of PPARγ and the active site of RXRα. 

In the RXRα-PPARγ system, the correlated motions of RXRα are present through DBD–hinge, hinge–H1, H2–H4/5, H7–H9 and H10/11–H12 (Figure 8). Among these correlated motions, L436 in H10/11 interacts directly with 9CR (Figure 3D). E219 in the hinge and K356 at H7 form protein–protein interactions with R153 in the DBD and E407 in H9 of PPARγ, respectively (Appendix A). 

PPARγ in the RXRα-PPARγ system shows that the correlated motions take place through residues in hinge–H1, H3–H4/5, H4/5–H7, H9–H10/11 loop, and H10/11–H12 (Figure 8). Of these correlated motions, only G361 and D362 in H7 are close to two residues (F363 and M364) that interact directly with RZG (Figure 3C,E). Only two residues (Q444 and T447) in H10/11 of PPARγ form protein–protein interactions with two residues (R426 and S427) in H10/11 of RXRα (Appendix A). In the interdomain of the RXRα-PPARγ system, there is only one interaction between T449 in H12 of RXRα and Q187 in the hinge of PPARγ. 

## 3. Discussion

Experimental data collected with X-ray crystallography, electron microscopy, and SAXS have provided structural insight regarding the active and inactive states of the RXRα-PPARγ system. In the active state, the RXRα-PPAR system is coupled to activating ligands and coactivators, which facilitate DNA binding, activating gene expression [27,28,29]. Previous MD simulations at the nanosecond timescale (several replicates of 100 ns each) using the activated state of the RXRα-PPARγ system with DNA bound revealed a flux of correlated motions coming from DNA to the LBDs, with important participation of PPARγ-DBD compared with RXRα [35], results in agreement with hydrogen/deuterium exchange experiments of the permissive VDR–RXR complex [30]. In this study, we explored the structural and energetic basis of the molecular recognition between the active RXRα-PPAR system and DNA at the microsecond timescale. We employed the structure of RXRα-PPAR co-crystallized with their agonists and coactivators and forming or not forming a complex with DNA through its DBD (PDB entry 3DZY). 

Clustering studies over the equilibrated simulation time revealed that the main regions contributing to the protein–ligand complex are similar for RZG in the RXRα-PPARγ and RXRα-PPARγ-DNA systems; the only notable differences relate to stabilization of the TZD headgroup, whose stabilization through hydrogen bonds was more like the crystallographic structure [25] in the RXRα-PPARγ system. In contrast, 9CR in the RXRα-PPARγ system is stabilized by more interactions in the LBD compared with the RXRα-PPARγ-DNA system, including H6, H7, and H10/11 by part of the RXRα-PPARγ system. Protein–protein interactions revealed more interactions for the RXRα-PPARγ than the RXRα-PPARγ-DNA system, mostly through the DBD and the hinge of RXRα with residues of the LBD of PPARγ. In contrast, in the RXRα-PPARγ-DNA system, most of the interactions are through the LBD of both receptors.

Binding free energy calculations of protein–ligand interactions using the MMGBSA approach revealed that the affinity of RZG is similar in both systems. In contrast, the affinity of 9CR is higher in the RXRα-PPARγ-DNA system. These differences suggest RXRα plays an important role in modulating molecular recognition in the system. Binding free energy calculations of protein–protein interactions using the MMGBSA approach showed that coupling DNA to the RXRα-PPARγ heterodimer contributes to decrease the affinity between both partners, a finding consistent with the different protein–protein interactions observed in the two systems.

Based on per-residue free energy decomposition analysis of the two systems, RZG is better stabilized in the RXRα-PPARγ-DNA system via more residues that contribute to the affinity. Similar analysis for 9CR showed that this ligand is better stabilized in the RXRα-PPARγ-DNA system, and the residues participating in stabilization are different in the two systems, indicating that DNA binding by the RXRα-PPARγ system significantly impacts molecular recognition of RZG and 9CR at the PPARγ and RXRα binding sites, respectively, but more importantly for the latter system. 

Per-residue free energy decomposition for the protein–protein interface revealed greater participation of RXRα residues (in the DBD, H9, and H10/11) than PPARγ residues (in the DBD, β2–β4, H6, H7, H8-H9 loop, and H10/11) in the RXRα-PPARγ system. For the RXRα-PPARγ-DNA system, the RXRα residues involved are also more important in recognition and are located in similar regions to those in the RXRα-PPARγ system. By contrast, the PPARγ residues involved are different than for the RXRα-PPARγ system, located in β2–β4, H8–H9 loop, H9, H10/11, and H12. Overall, PPARγ is more important in regulation of the system regardless of whether DNA is bound. These results suggest that PPARγ has a greater role in regulation of the free and bound DNA state, a finding in line with previous MD simulations using the RXRα-PPARγ-DNA system [35].

PCA and the FEL indicated that the RXRα-PPARγ system explores a larger conformational space and more major stable conformations than the RXRα-PPARγ-DNA system. Hence, DNA binding by the RXRα-PPARγ complex reduces the conformational entropy and decreases the number of major stable conformational ensembles, but thermostable conformations persist, result in line with experimental findings revealing that DNA binding propagates conformational changes that stabilize the heterodimer [40]. Projection along PC1 and PC2 showed that RXRα-PPARγ system displayed higher collective motions along DBD and H2’-H3 loop of PPARγ. The motion at H2’-H3 loop exhibited a conformational behavior different for free and bound DNA systems. For free system, the H2’-H3 loop displayed a sliding motion, strongly associated to PPARγ surface and into the distance to form interactions with H6-H7 loop, results are somewhat in agreement with previous reports for phosphorylated PPARγ within RXRα-PPARγ-DNA system [41]. In contrast, a flapping motion over the ligand binding site of LBD was observed for the H2’-H3 loop in RXRα-PPARγ-DNA complex, a similar result as previously reported for the ternary RXRα-PPARγ-DNA complex [41]. In general, it can be appreciated that the conformational dynamic is more affected in PPARγ than RXRα upon DNA binding; this observation is consistent with Hydrogen/Deuterium Exchange research on the VDR-RXR complex, which showed that RXR-DBD dynamics is less impacted by DNA binding as compared to its partner receptor [30].

Correlated motions among different regions of RXRα or PPARγ in the RXRα-PPARγ-DNA system indicate that the other parts of LBD have their dynamics connected to DBD in the case of PPARγ, but no RXRα, effect in connection with that reported elsewhere [35]. Comparison of the intradomain correlated motions of RXRα-PPARγ and RXRα-PPARγ-DNA system denotes a stronger correlation map for RXRα-PPARγ-DNA, supporting the hypothesis that there is a flux of conformational information emerging from DNA to the LBDs [35], which is facilitated by the PPARγ-DBD, since the communication between DBD-hinge-LBD was only observed for PPARγ.

## 4. Material and Methods

### 4.1. System Setup

The initial coordinate of the PPARγ-RXRα-DNA complex was taken from crystallographic structures placed in the Protein Data Bank (PDB) with the PDB entry 3DZY [25]. The structure contains PPARγ isoform 1, RXRα, coactivator peptides bound to both PPARγ and RXRα, 9CR bound to RXRα, and RZG bound to PPARγ. The missing portions of RXRα (residues 242–263) and PPARγ (residues 260–275) were constructed by using Modeller 9.17 [42], RXRα (Uniprot entry P19793) and PPARγ isoform 1 (Uniprot entry P37231) sequences and chain A and D from PDB entry 3DZY. In PPARγ-RXRα and PPARγ-RXRα-DNA system, the coactivator peptides were considered. The RXRα-PPARγ system was constructed by deleting DNA from the constructed RXRα-PPARγ system. The quality of the systems was evaluated with MolProbity [43], which reported the following residues values in preferred regions of the Ramachandran plot: 155/160 (96.87%). Protonation states of ionizable groups were predicted with the server PROPKA at neutral pH [44]. Complete systems comprised approximately 159,000 and 156,000 atoms for PPARγ-RXRα-DNA and PPARγ-RXRα systems, respectively. 

### 4.2. MD Simulations

The RXRα-PPARγ-DNA, and RXRα-PPARγ systems forming a complex with 9CR and RGZ were solvated with the TIP3P water model [45] in a periodic cubic box with a solute-box wall distance of 1.5 nm. MD simulations were run using Amber 16 software [46]. The net charges of each system were neutralized with Na^+^ and Cl^−^ counterions at the physiological concentration of 0.15 M. The protein, water molecules, and ions were specified by the ff14SB AMBER force field [47], and the DNA was specified by the ff99bsc0+OL15 force field [48]. The ligand force field was built by designating AM1-BCC atomic charges using the general Amber force field (GAFF) [49]. The minimization and relaxing protocol consisted of 1000 steps of energy minimization; 1000 ps of pre-relaxation with the protein-heavy and ligand atoms restrained with a harmonic restraint weight of 10 kcal mol^−1^ Å^−2^; 1000 ps of relaxation without restrictions on the ligand atoms; 1000 ps of relaxation with no restrictions to the side chains of the residues 5 Å around the ligand; and 1000 ps of NPT relaxation without restrictions for protein and ligand atoms. MD simulations consisted of three independent 1 µs simulations, each initialized from the same minimized geometry with distinct initial atomic velocities determined from a Maxwell distribution at 310 K. In the production MD simulations, the SHAKE algorithm [50] was selected to restrain all bonds to equilibrium lengths, allowing a time step of 2 fs. The particle-mesh Ewald (PME) method [51] was used to treat the long-range electrostatic interactions with a cut-off of 1.0 nm. Van der Waals interactions were treated with the Verlet scheme with a cut-off distance of 1.0 nm. The system temperature was controlled using the weak-coupling algorithm [52] at 310 K with a time constant of 0.1 ps, and the system pressure was kept at 1 atm using a Berendsen barostat.

### 4.3. MD Trajectory Analysis

For the triplicate MD simulations, the time-dependent Cα root-mean-squared deviation (RMSD), the radius of gyration (Rg), and clustering analysis were evaluated using AmberTools16. For each complex, the equilibrated part of each simulation was concatenated into a single joined trajectory, based on which PCA and clustering analysis were developed by using AmberTools16 (considering the Cα atoms). Most populated conformations were found through a cluster analysis using the kclust algorithm in the MMTSB toolset (http://www.mmtsb.org/software/mmtsbtoolset.html (accessed on 10 April 2022)) and considering a cut-off of 3.5 Å. PCA was calculated from the diagonalization of the covariance matrix [53]. This analysis provided the two extreme projections along the first and second eigenvectors. The first two eigenvectors were considered the reaction coordinates to construct the two-dimensional FEL based on PCA as reported elsewhere [54]. A correlation map was constructed by using dynamical cross-correlation motion [55]. This analysis allows to identify correlated motions occurring among neighboring residues that form part of a secondary structure element or distant residues localized in different regions or domains. For PCA and correlation map analysis, the DNA was not considered. The protein-protein interactions at the heterodimeric RXRα-PPARγ, and RXRα-PPARγ-DNA interfaces was explored by Molecular Operating Environment MOE 2014.0901. Figures were built by using PyMOL [26].

#### Binding Free Energy Calculations

The binding free energies of protein–ligand and protein–protein interactions were evaluated by using the MMGBSA approach [56] implemented in AmberTools16. MMGBSA is an end-point approach capable of obtaining the binding free energy (*ΔG_bind_*) for protein-ligand and protein-protein complexes based purely on the ensemble of the bound complexes without contemplating either the physical or the non-physical intermediates [57]. The binding free energy was estimated over the last 0.4 µs of the equilibrated simulation time concatenated into a single joined trajectory, saving 4000 representative conformations. The solvation free energy was determined by using implicit solvent models [58], ionic strength of 0.15 M, and solute and solvent dielectric constants of 4 and 80, respectively. The binding free energies were estimated as described elsewhere [59]. 

## 5. Conclusions

We used structural data and MD simulations coupled with the MMGBSA approach, PCA, the FEL, and correlated motion analyses to explore the structural and energetic differences between the RXRα-PPARγ and RXRα-PPARγ-DNA systems. Clustering and energetic analysis using the MMGBSA approach indicated that 9CR experiences more significant changes in affinity and map of interactions, whereas RZG shows some small differences. The affinity of the RXRα-PPARγ system is reduced upon DNA binding. PCA and FEL studies revealed that DNA binding by the RXRα-PPARγ system reduces the conformational entropy, but thermostable conformations persist in both systems. There are correlated motions between the hinge and the LBD for RXRα or PPARγ forming part of the RXRα-PPARγ, and RXRα-PPARγ-DNA systems, but the motion between DBD–hinge–LBD only occurs for PPARγ within RXRα-PPARγ-DNA system, suggesting a greater role of PPARγ in regulation of the free and bound DNA state, in line with previous reports. Projection along PC1 and PC2 demonstrated an increase in the mobility of H2’-H3 loop of PPARγ within RXRα-PPARγ-DNA system, linked to a flapping motion of this loop over the ligand binding site that at the time promotes better interactions with RZG. H11-H12 loop of RXRα within RXRα-PPARγ-DNA system experiences a reduction in mobility that contributes to increase affinity by 9CR. Taken together, we conclude that DNA binding by the RXRα-PPARγ system contributes to stabilize the heterodimer through a series of conformational changes that improve the map of interactions at the ligand binding site of LBD of both protomers of RXRα-PPARγ system. Based on this insight, we theorize that when designing new PPARγ agonists, it may be favorable to evaluate the structural and energetic changes and compare them with known PPARγ agonists.

## Figures and Tables

**Figure 1 molecules-27-05778-f001:**
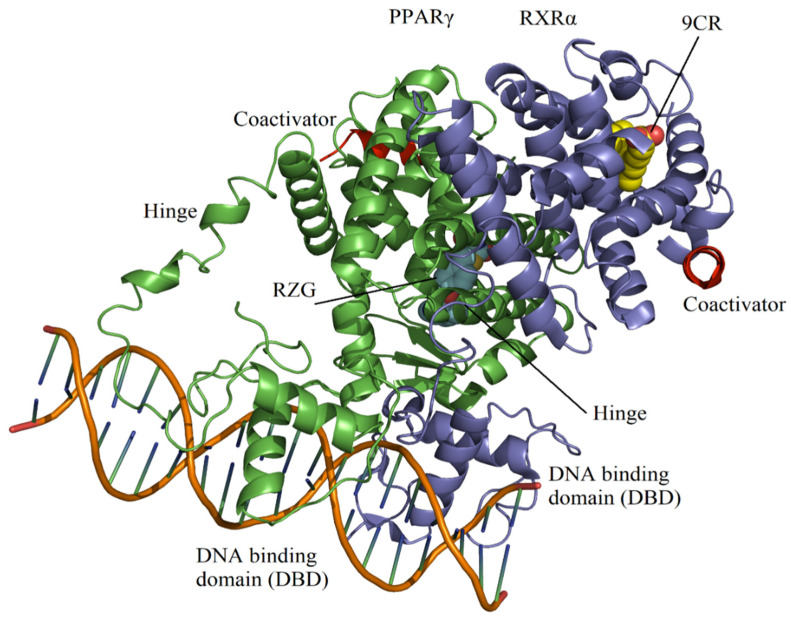
Structural topology of the RXRα-PPARγ-DNA system (PDB entry 3DZY) [25]. PPARγ and RXRα are presented in green and purple, respectively. For each receptor, a flexible hinge connects the DBD and the LBD. Figure was built by using PyMOL [26].

**Figure 2 molecules-27-05778-f002:**
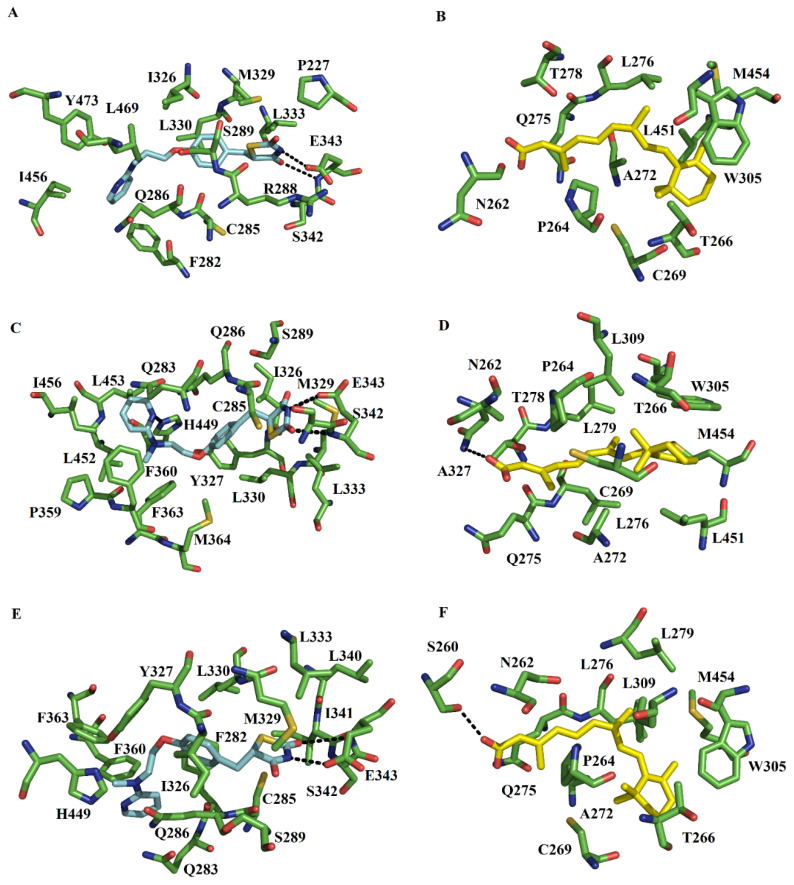
Protein-ligand interactions of RGZ and 9CR in the PPARγ-RXRα-DNA system. Interactions with RGZ and 9CR at the ligand-binding site of PPARγ and RXRα, respectively, are present in the first [RGZ (**A**) and 9CR (**B**)], second [RGZ (**C**) and 9CR (**D**)], and third [RGZ (**E**) and 9CR (**F**)] most populated conformations. These structures are representative from the three major clusters. Arrows indicate HBs.

**Figure 3 molecules-27-05778-f003:**
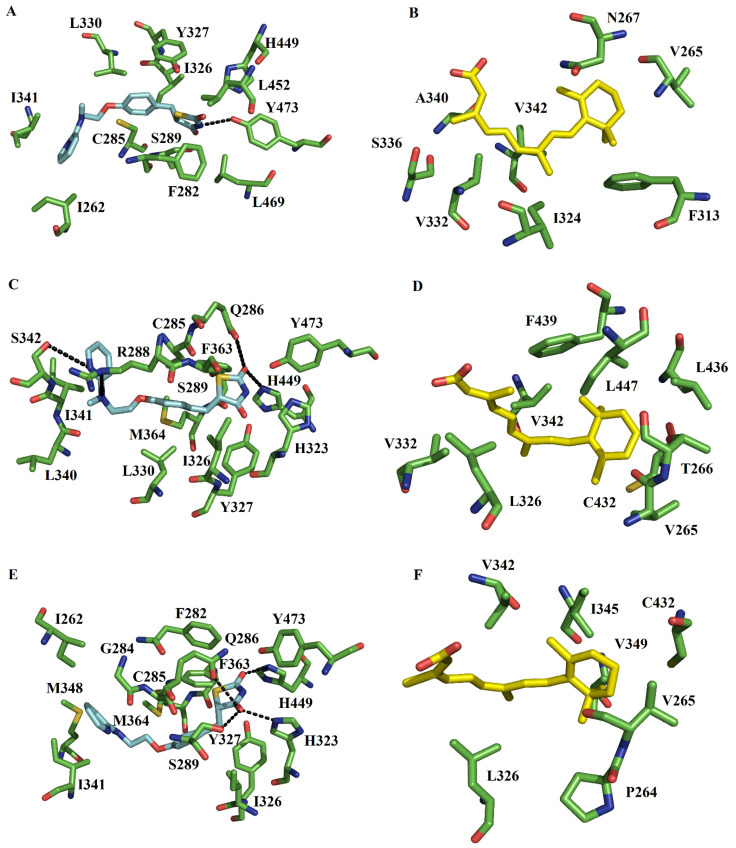
Map of interactions of RGZ and 9CR in the PPARγ-RXRα system. Interactions of RGZ and 9CR at the ligand-binding site of PPARγ and RXRα, respectively, are present in the first [RGZ (**A**) and 9CR (**B**)], second [RGZ (**C**) and 9CR (**D**)], and third [RGZ (**E**) and 9CR (**F**)] most populated conformations.

**Figure 4 molecules-27-05778-f004:**
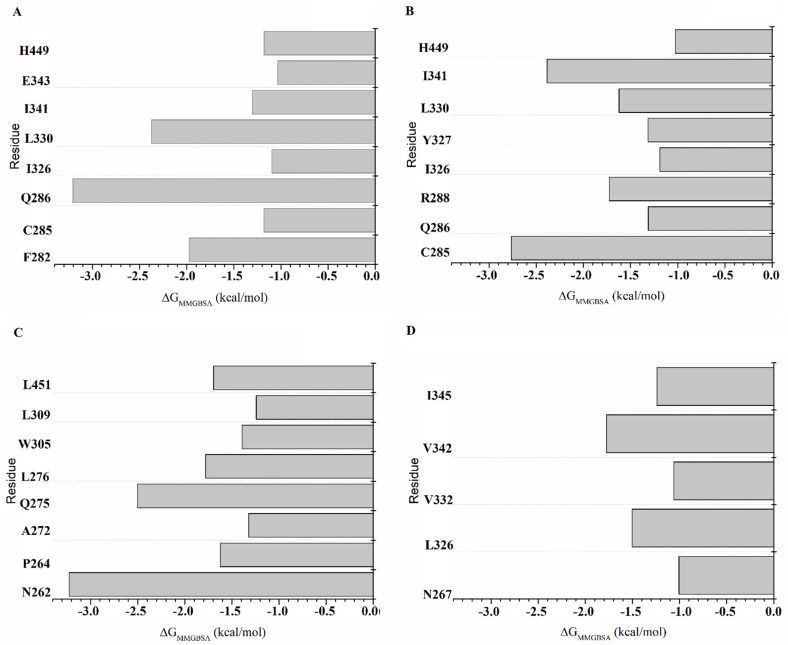
Decomposition of the binding free energy(*ΔG_MMGBSA_*) for protein-ligand interactions of the RXRα-PPARγ-DNA, and RXRα-PPARγ systems coupled to RGZ and 9CR. Per-residue free energy of RGZ (**A**) and 9CR (**C**) in the PPARγ-RXRα-DNA system. Free energy of RGZ (**B**) and 9CR (**D**) in the PPARγ-RXRα system.

**Figure 5 molecules-27-05778-f005:**
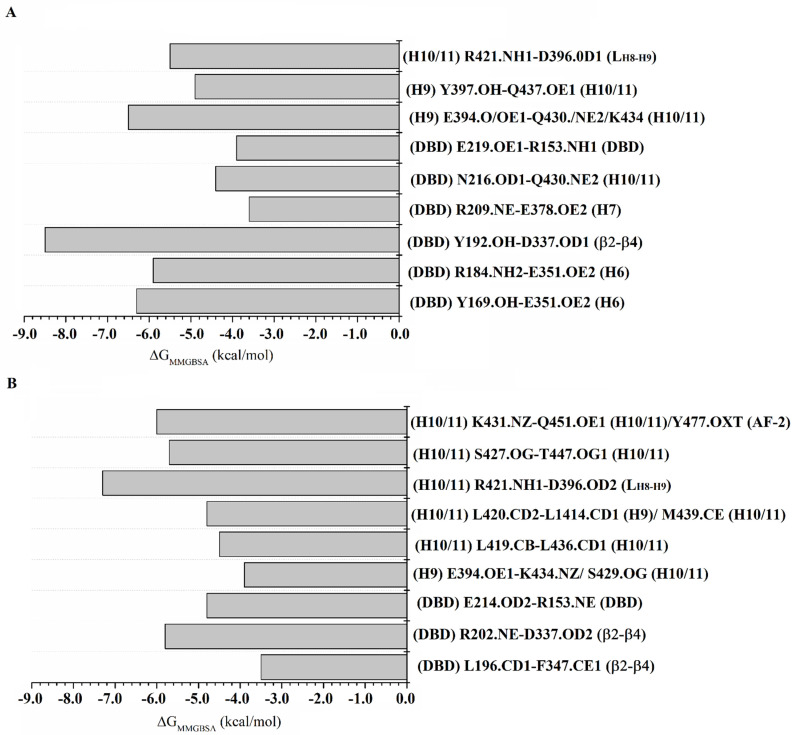
Per-residue free energy for protein-protein interactions of the RXRα-PPARγ (**A**), and RXRα-PPARγ-DNA (**B**) systems coupled to RZG and 9CR (values are in Kcal/mol). Residues at the left and right of the interaction correspond to RXRα and PPARγ, respectively.

**Figure 6 molecules-27-05778-f006:**
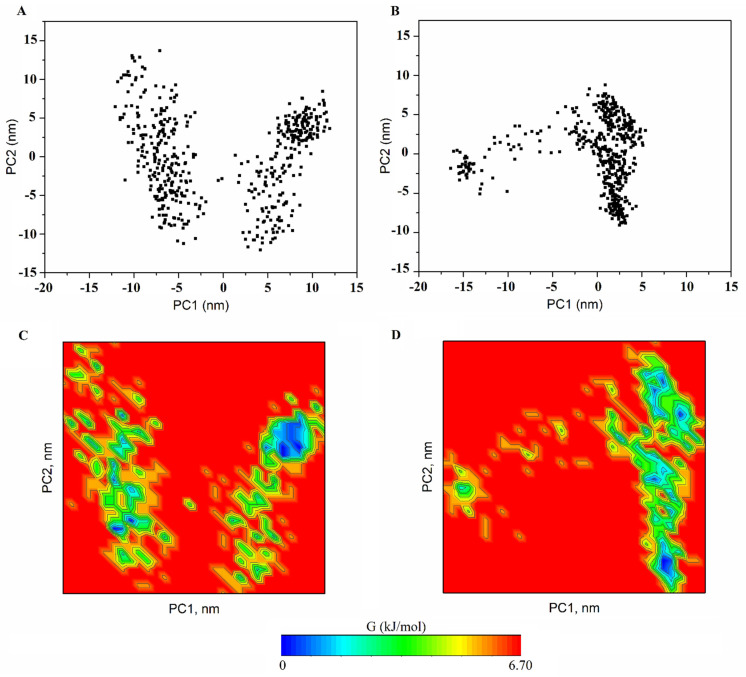
Projection in phase space and FELs of the two systems. FELs as a function of the projection of PC1 and PC2 onto the essential subspace of the PPARγ-RXRα system (**A**) and the PPARγ-RXRα-DNA system (**B**). Projection of the motion in the phase space along the first and second eigenvectors (PC2 versus PC1) for the PPARγ-RXRα system (**C**) and the PPARγ-RXRα-DNA system (**D**). The color bar represents the relative free energy value in kcal/mol.

**Figure 7 molecules-27-05778-f007:**
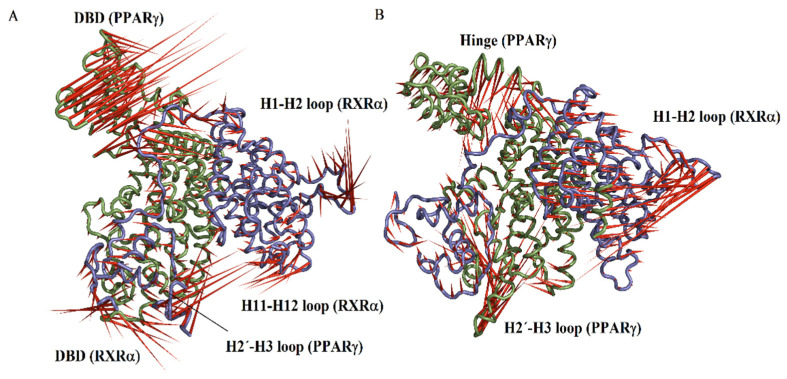
Graphic representation of the projection on PPARγ-RXRα and PPARγ-RXRα-DNA systems. Graphical depiction of the two extreme projections along PC1 vs. PC2 for the PPARγ-RXRα (**A**) and PPARγ-RXRα-DNA (**B**) systems. The direction and magnitude of movements are described as porcupine drawings.

**Figure 8 molecules-27-05778-f008:**
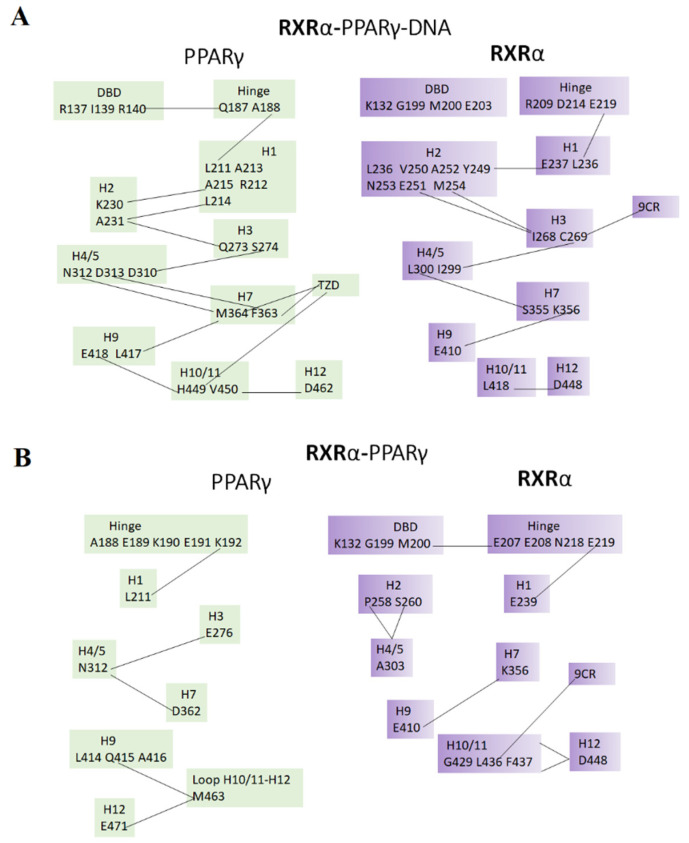
Intradomain correlated motions of RXRα-PPARγ-DNA (**A**), and RXRα-PPARγ (**B**) systems. Black lines denote correlated motions between domains.

**Table 1 molecules-27-05778-t001:** Binding free energy components for protein-ligand and protein-protein interactions of the RXRα-PPARγ, and RXRα-PPARγ-DNA systems are calculated by using MMGBSA approach (values are in Kcal/mol).

*System*	*ΔE_vdw_*	*ΔE_ele_*	*ΔG_ele,sol_*	*ΔG_npol,sol_*	*ΔG_mmgbsa_*
Protein-ligand					
PPARγ_RSG_-RXRα_9CR_					
PPARγ_RZG_	−48.65 ± 3.0	−18.47 ± 6.5	32.82 ± 4.0	−6.56 ± 0.2	−40.86 ± 5.0
RXRα_9CR_	−38.94 ± 3.0	43.78 ± 10.0	−29.38 ± 9.9	−5.49 ± 0.3	−30.04 ± 3.0
PPARγ_9CR_-RXRα_9CR_-DNA					
PPARγ_RZG_	−51.33 ± 3.0	−17.88 ± 6.0	33.23 ± 3.0	−6.80 ± 0.3	−42.78 ± 5.0
RXRα_9CR_	−45.99 ± 2.5	211.83 ± 13.0	−198.96 ± 12.0	−6.41 ± 0.2	−39.43 ± 4.0
Protein-protein					
PPARγ_RZG_-RXRα_9CR_	−277.48 ± 15.0	−1245.50 ± 133.5	1446.38 ± 133.0	−42.98 ± 2.3	−119.59 ± 18.0
PPARγ_RZG_-RXRα_9CR_-DNA	−242.90 ± 12.0	−750.53 ± 110.0	935.68 ± 107.0	−37.09 ± 1.8	−94.84 ± 13.0

## Data Availability

The datasets supporting the conclusions of this research are contained within the paper and its additional files.

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
