# Peer review of "Microsecond MD Simulations to Explore the Structural and Energetic Differences between the Human RXRα-PPARγ vs. RXRα-PPARγ-DNA"

_molecules, 2022, doi:10.3390/molecules27185778_

Round 1

Reviewer 1 Report

In the abstract, the authors refer to X-ray and MD. What type of X-ray experiment? Nothing is mentioned in the main text about X-ray study.
The authors provide a theoretical study, but the title implies otherwise. The title should be in line with the content.
Author contributions should be stated.
The manuscript does not lie within the scope the journal particularly the medicinal chemistry section.

Author Response

Response to reviewers

 We want to thank reviewers for their fruitful comments and observations that, without hesitation, will contribute to improving the quality of this scientific contribution.

Reviewer 1

In the abstract, the authors refer to X-ray and MD. What type of X-ray experiment? Nothing is mentioned in the main text about X-ray study.
The authors provide a theoretical study, but the title implies otherwise. The title should be in line with the content.
Author contributions should be stated.
The manuscript does not lie within the scope the journal particularly the medicinal chemistry section.

Response:

We have modified the abstract to avoid confusion with respect to X-ray experiments.

The title has been modified to be more in line with the content.

The author contributions have been stated.

Reviewer 2 Report

The manuscript is well organized. The results are compared with previous studies.

The active sites (MD result) must be compared in detail with the literature.

Figures 2-3 and S3 should be corrected, it is not clear.

Author Response

Response to reviewers

 We want to thank reviewers for their fruitful comments and observations that, without hesitation, will contribute to improving the quality of this scientific contribution.

Reviewer 2

The manuscript is well organized. The results are compared with previous studies.

The active sites (MD result) must be compared in detail with the literature.

Figures 2-3 and S3 should be corrected, it is not clear.

Response:

We have cited more information comparing the active sites with the literature. Figures 2-3 and S3 have been improved.

Reviewer 3 Report

The manuscript molecules-1882142 claims the Structural and energetic differences between human 2 RXRα–PPARγ heterodimer in presence/absence of DNA. But there are lots of research papers published on this topic with experimental validation. So what is the novelty in this work? Why the authors have not proposed any agonists using docking and other in silico validations? The image captions and qualities are not clear and difficult to understand. Minimization steps should be described in the MD simulations protocol. Authors have used very long range MD simulation study (i.e. 1000 ns) to evaluate the stability of the protein-ligand complexes. No doubt the long range simulation used here is quite appreciable to study such macromolecular dynamic characteristics. Indeed, authors have generated huge amount of data through MD simulation. However, in the manuscript, the presented MD simulation analysis was not performed rigorously. Still, basic form of analysis is missing. Like RoG and ligand RMSF, and SASA parameters should be incorporated and discussed.

Author Response

Response to reviewers

 We want to thank reviewers for their fruitful comments and observations that, without hesitation, will contribute to improving the quality of this scientific contribution.

Reviewer 3

The manuscript molecules-1882142 claims the Structural and energetic differences between human 2 RXRα–PPARγ heterodimer in presence/absence of DNA. But there are lots of research papers published on this topic with experimental validation. So what is the novelty in this work? Why the authors have not proposed any agonists using docking and other in silico validations? The image captions and qualities are not clear and difficult to understand. Minimization steps should be described in the MD simulations protocol. Authors have used very long range MD simulation study (i.e. 1000 ns) to evaluate the stability of the protein-ligand complexes. No doubt the long range simulation used here is quite appreciable to study such macromolecular dynamic characteristics. Indeed, authors have generated huge amount of data through MD simulation. However, in the manuscript, the presented MD simulation analysis was not performed rigorously. Still, basic form of analysis is missing. Like RoG and ligand RMSF, and SASA parameters should be incorporated and discussed.

Response:

Although there are several articles on this topic with experimental validation, our research provides new information about the structural and energetic impact of the coupling of DNA with RXRα–PPARγ heterodimer. Our study offers novelty information about the differences in the correlated motions, being more cooperative when the DNA is forming a complex with the RXRα–PPARγ heterodimer, which at the time impacts the protein-protein interactions and protein-ligand affinity. In addition, our results also corroborated the more significant role of PPARγ in the regulation of the free and bound DNA state.

Although we would have wanted to explore other ligands combining docking and MD simulations, in this research, we only decided to use ligands with experimental information since docking calculations sometimes provide misleading information about the protein-ligand interactions.

The image captions and qualities have been modified.

Information on minimization has been highlighted in the manuscript.

Information on RoG, SASA and RMSF analysis has been included in the manuscript.

Round 2

Reviewer 1 Report

The overall quality of the work is average and I can't find a sound significance to the medicinal chemists.

In general, the paper provides good in silico study and detailed analysis.

Reviewer 2 Report

Accept

Reviewer 3 Report

The authors have addressed the queries.